# Monitoring for Changes in Spring Phenology at Both Temporal and Spatial Scales Based on MODIS LST Data in South Korea

**Chi Hong Lim [1], Song Hie Jung [2], A Reum Kim [3], Nam Shin Kim [1] and Chang Seok Lee [4,*]**

[1] Division of Ecological Survey Research, National Institute of Ecology, 1210 Geumgang-no, Maseo-myeon, Seocheon 33657, Korea; sync03@nie.re.kr (C.H.L.); geotop@nie.re.kr (N.S.K.)

[2] Gwangneung Forest Conservation Center, Korea National Arboretum, 415 Gwangneungsumogwon-ro, Soheul-eup, Pocheon 11186, Korea; jung2024@nie.re.kr

[3] Graduate School of Seoul Women's University, Seoul Women's University, 621 Hwarang-no, Nowon-gu, Seoul 01797, Korea; dkfma@swu.ac.kr

[4] Division of Chemistry and Bio-Environmental Sciences, Seoul Women's University, Seoul Women's University, 621 Hwarang-no, Nowon-gu, Seoul 01797, Korea

\* Correspondence: leecs@swu.ac.kr

**Abstract:** This study aims to monitor spatiotemporal changes of spring phenology using the green-up start dates based on the accumulated growing degree days (AGDD) and the enhanced vegetation index (EVI), which were deducted from moderate resolution imaging spectroradiometer (MODIS) land surface temperature (LST) data. The green-up start dates were extracted from the MODIS-derived AGDD and EVI for 30 Mongolian oak (*Quercus mongolica* Fisch.) stands throughout South Korea. The relationship between green-up day of year needed to reach the AGDD threshold (DoY$_{AGDD}$) and air temperature was closely maintained in data in both MODIS image interpretation and from 93 meteorological stations. Leaf green-up dates of Mongolian oak based on the AGDD threshold obtained from the records measured at five meteorological stations during the last century showed the same trend as the result of cherry observed visibly. Extrapolating the results, the spring onset of Mongolian oak and cherry has become earlier (14.5 ± 4.3 and 10.7 ± 3.6 days, respectively) with the rise of air temperature over the last century. The temperature in urban areas was consistently higher than that in the forest and the rural areas and the result was reflected on the vegetation phenology. Our study expanded the scale of the study on spring vegetation phenology spatiotemporally by combining satellite images with meteorological data. We expect our findings could be used to predict long-term changes in ecosystems due to climate change.

**Keywords:** AGDD; climate change; EVI; MODIS LST; Mongolian oak; phenology

## 1. Introduction

Global climate change can lead to meaningful changes in plant phenology as temperature affects the timing of development, not only alone but also by interacting with other factors, such as the photoperiod [1,2]. A number of recent studies showed that the growing season of vegetation has been extended by recent climate change [3–9] and this extension mostly results from the earlier onset of spring. Many of those studies have reported a correlation between earlier spring phenology and rising temperature but have showed different effects on the end of the growing season [1,10–15]. Therefore, spring phenology is the most indisputable monitoring tools for the seasonality of plant species [5,13,16–18]. Spring phenology is thought to greatly affect the productivity and carbon budget

of temperate and boreal ecosystems [17,19]. In addition, difference among species in spring phenology is a significant mechanism for maintaining the coexistence of species in diverse plant communities by reducing competition for pollinators and other resources [1,20,21].

Earlier spring green-up due to climate change can increase risk of a false spring, when subsequent hard freezes damage new, vulnerable plant growth in ecological and agricultural systems [22–24]. Therefore, the climate-driven changes in the phenology of plant species are not only the simple change of phenomena but also the change of ecological functions. For example, phenological mismatches due to the time shifts might occur when organisms that typically interact, such as plants and pollinators, are no longer active at the same time [25,26]. According to Kudo and Ida [27], although both the onset of flowering and the first appearance of overwintered queen bees in *Corydalis ambigua* populations were closely related to snowmelt time and/or spring temperature, flowering tended to be ahead of the first pollinator appearance when spring came early, resulting in lower seed production due to low pollination services.

Land use pattern can account for much of the temperature variation [28]. In particular, urbanization is increasingly having an important anthropogenic impact on climate and has significantly influenced terrestrial ecosystems [29–32]. It can modify the local climate on daily, seasonal, and annual scales [33–36]. The changes in climate due to intensive land-use by humankind in urban areas can therefore be regarded as a kind of climate change on a local scale. The change in the local climate as a result of a distinguishing land-use pattern in an urban area is often referred as the urban heat island (UHI) effect. The UHI effect is recognized as being caused by a reduction in the latent heat flux and an increase in sensible heat in urban areas as vegetated and evaporating soil surfaces are replaced by pavement and building materials with relatively impervious and low albedo [37–40]. Consequently, cities are exposed to climate change from greenhouse gas-induced radiative forcing, and localized effects from urbanization such as the urban heat island. Warming and extreme heat events due to urbanization and increased energy consumption are simulated to be as large as the impact of doubled $CO_2$ in some regions [41].

Numerous techniques to observe how phenology has changed, including ground-based observations [6,42–44], digital repeat photography [45–49], and satellite remote sensing [7,16,23,50–52] have been developed. Among these techniques, satellite remote sensing is taking the spotlight because of the advantage of providing multi-decadal records of vegetation phenology across larger spatial scales than other techniques [53–55]. In particular, advances in both temporal and spatial resolutions and ease of availability in recent years have made remotely sensed data derived from moderate resolution imaging spectroradiometer (MODIS) the most generally used datasets in phenological observation at the extensive scale such as regional and global levels [52,56–58]. In addition, as novel algorithms are developed to resolve the critical limitations of the satellite measurement such as cloud contamination and cloud shadow effect [57,59–67].

This study was carried out to monitor changes of vegetation phenology due to climate change. To carry out this study, we established the following hypotheses: First, the green-up date of the Mongolian oak will respond to the accumulative value of the temperature at which the physiological activity of the plant is initiated. Second, the green-up date of the Mongolian oak will be spatially different depending on local climate conditions. Third, the green-up date of the Mongolian oak will be advanced according to climate change. Fourth, urbanization will bring about a change in the green-up date of the Mongolian oak.

To verify these hypotheses, we analyzed the relationship between the temperature and the vegetation phenology data obtained from MODIS images for 30 representative Mongolian oak forest sites selected based on the national inventory data for the natural environment that the Korean government constructed. To verify this data, we analyzed the relationship between day of year (DoY; 159 accumulated growing degree days (AGDD)) and average temperature based on weather data from all 93 meteorological stations in South Korea. To trace the temporal variation of phenology that the Mongolian oak showed over the past century, we traced the change in DoY (159 AGDD) based on weather data from the five major cities that have been measuring weather data for the longest period in Korea. On the other hand, we analyzed the cherry blossom date data of five major cities in

Korea, which have been surveying cherry blossom dates for a long time to re-validate the temporal variation of the green-up date of the Mongolian oak deducted from the temperature data. Finally, we analyzed the spatial variation of temperature anomaly throughout South Korea and by land-use type of urban, rural, and forest areas.

## 2. Materials and Methods

### 2.1. Study Area

In the study of vegetation phenology, it is necessary to select pure stands to reduce errors from differences in phenology according to plant species. Mongolian oak (*Quercus mongolica* Fisch.) forest, which dominates the temperate deciduous broadleaved forest zone, was selected as the target vegetation for this study (Figure 1). The phenological signal of canopy reflectance from deciduous broadleaved forests can be clearly defined, ensuring accurate measurement of the change in growing season. Therefore, we selected Mongolian oak (*Quercus mongolica* Fisch.), which is not only a dominant species in the temperate deciduous broadleaved forest zone but also the representative species forming the late successional forest on the Korean Peninsula, as the target species of this study. Among the species belonging to the *Quercus* genus, *Q. mongolica* grows at the highest elevation and thus, would likely be a species sensitive to global warming. Mongolian oak begins leaf unfolding and senescence the earliest among deciduous oaks growing in Korea. Therefore, it is estimated that Mongolian oak could be the best species to derive phenological transition date from MODIS image with low spatial resolution.

In order to verify the green-up date of Mongolian oak deducted from the temperature data obtained from MODIS image of 1 km spatial resolution, meteorological data were collected from all (93) meteorological stations in South Korea. Moreover, meteorological data were collected from five major cities that have been measuring weather data for the longest period in Korea to deduce the change of green-up dates of Mongolian oak over year. Furthermore, cherry blossom data were collected from the same cities that have been observing flowering of the cherry visibly for the longest period in Korea to verify the change of green-up dates of Mongolian oak deducted from temperature variation over the year.

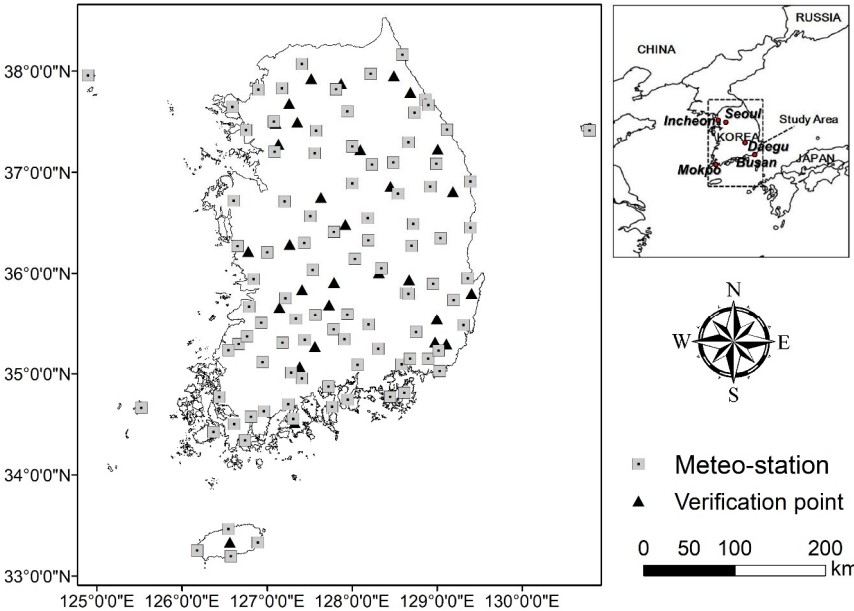

**Figure 1.** Map of the study area and spatial distribution of the primary sampling points. Meteo-station: Meteorological stations; Verification point: Mongolian oak (*Quercus mongolica* Fisch.) forests selected for validation.

*2.2. Experimental Design*

The overall procedure of the experiments was shown in Figure 2. We selected 30 representative Mongolian oak forest sites based on the national inventory data for the natural environment that the Korean government constructed. The Mongolian oak forest is a typical late successional forest of the temperate deciduous broadleaved forest, and thus the forest is located relatively far from the human living environment. Therefore, it is very difficult to obtain weather data from the site where this forest was formed and, thereby, we obtained temperature data by analyzing MODIS image data. We deducted DoY (159 AGDD), the period required to reach AGDD 159 °C necessary for the leaf unfolding of the Mongolian oak obtained through previous research (Lim et al. [60]) based on the data and clarified the relationship between DoY (159 AGDD) and the average temperature.

To verify this data, we analyzed the relationship between DoY (159 AGDD) and the average temperature based on weather data from 93 stations in Korea. Based on these results, we tracked the temporal variation of phenology that the Mongolian oak showed over the past century. These data were re-validated through the analysis on the cherry blossom date data of five major cities in Korea, which have been surveying cherry blossom dates for a long time.

To enhance the reliability of these results, we added maps including isopleth of DoY for AGDD 159, 2015, elevation, and the land-use type and geographical and topographical information of each verification site as Appendix A.

*2.3. Data Collection and Pre-Processing*

The details for the remotely sensed available data sources by station are listed in Table 1. To deduct spring green-up timing of Mongolian oak, we used an 8-d interval air temperature map dataset (1 km spatial resolution, root mean squared error: 3.91) in South Korea from January 1 to June 26 in 2015, which was reconstructed by Lim et al. [60] using MODIS land surface temperature (LST) imagery. In addition, we collected the MODIS surface reflectance product (MOD09GA) from January 1 to June 26 in 2000, 2005, 2010, and 2015 to deduct the spring green-up date based on the enhanced vegetation index (EVI).

Daily air temperature data (minimum and maximum temperature) in meteorological stations from January 1 2015 to July 2 2015 were obtained from KMA (Korea Meteorological Administration). In addition, daily mean air temperature data from January 1 1907 to December 31 2014 and the first flowering date of cherry (*Prunus serrulata* Lindl.) from 1921 to 2014 from five meteorological stations were acquired from KMA (Table 1).

To analyze the relationship between characteristics of air temperature and land-use intensity, a MODIS land cover product (MCD12Q1.005) was downloaded from the Earthdata website of NASA (now available at https://ladsweb.modaps.eosdis.nasa.gov/). The MODIS re-projection tool software (MRT V4.1) was used to re-project from the original Sinusoidal (SIN) to a TIFF file using the Universal Transverse Mercator projection (UTM Zone 52N, WGS84 ellipsoid). From the image, we extracted the pixels labeled by urban, rural, and forest land-use types in the MODIS land-cover type product (MCD12Q1). In addition, all water pixels were extracted for masking.

**Table 1.** A summary for data used in the study.

| | Data type | Original Resolution | Time Period | Utility | Source |
|---|---|---|---|---|---|
| MODIS product | Air temperature dataset | 8 d, 1000 m | 1/1/2015~6/26/2015 | Deduction of spring green-up date based on AGDD for Mongolian oak | Lim et al. [60] |
| | MOD09A1 | 8 d, 500 m | 1/1/2000~6/26/2000 1/1/2005~6/26/2005 1/1/2010~6/26/2010 1/1/2015~6/26/2015 | Deduction of spring green-up date based on EVI for Mongolian oak | NASA Earthdata website |
| | MCD12Q1 | Yearly, 500 m | 2015 | Land-use classification | |

| Meteorological station data | First flowering date of cherry | Yearly | 1921~2014 | Deduction of spring green-up date for cherry | Korea Meteorological Administration |
|---|---|---|---|---|---|
| | Air temperature record (maximum, minimum, mean) | Daily | 1/1/1907~12/31/2014 | AGDD calculation to extract phenological ascending trend | |
| | | | 1/1/2015~7/2/2015 | Deduction of relationship between phenological event dates based on AGDD and EVI | |

## 2.4. Indices Calculation

A series of indices that affect vegetation phenology, growing degree days (GDD; °C·d), and accumulated GDD (AGDD) were derived from the completely reconstructed 8-d air temperature maps. We used air temperature dataset by MODIS in the previous study of Lim et al. [60]. In addition, from the daily air temperature data, the AGDD at each station was calculated in the same way.

First, growing degree days (GDD; °C·d) were calculated using the equation from McMaster and Wilhelm [68]:

$$GDD_t = \frac{(T_{max \cdot t} + T_{min \cdot t})}{2} - T_{base} \tag{1}$$

where $T_{max \cdot t}$ and $T_{min \cdot t}$ are the maximum and minimum air temperatures at DoY$_t$, respectively, and $T_{base}$ is the temperature below which plant growth is zero. In this study, we set the base temperature as 5 °C. In addition, the GDD based on meteorological station data were estimated to validate the GDD derived from MODIS.

We accumulated 8-d interval GDDs by simple summation when the GDD exceeded the base temperature [3,57]:

$$GDD_{t-i} + i \times GDD_t \quad (GDD_t > T_{base}) \tag{2}$$

where $GDD_t$ is the 8-d mean GDD at DoY$_t$, and $i$ is the time interval coefficient (GDD from MODIS: 8; GDD from field measured data: 1), $AGDD_t$ is the GDDs accumulated from the beginning of the time period until DoY$_{t+7}$.

A study of Lim et al. [60] focused deduction of a meteorological indicator from reconstructed MODIS image and discussed applicability of the indicator. We carried out a study that actually applied the indicator. Furthermore, we extended the time range in conjunction with the weather data measured in 93 meteorological stations throughout the whole national territory, and we verified the estimated data compared to the phenology data of other plants, such as the cherry blossom date, which was recorded over a long period.

Based on 159 °C·d, the average of AGDD threshold values determined from field measurements when the spring green-up was started in Mongolian oak forests [60], a MODIS-derived AGDD threshold map was generated by counting pixelwise the number of DoY (hereafter, DoY$_{AGDD}$) required to reach the AGDD threshold to assess the timing of green-up of Mongolian oak throughout the whole national territory of South Korea.

To normalize the air temperature data, urban and water areas were masked. A linear regression model was fitted to the remaining air temperature with elevation and coordinates of X and Y for determining the constant and linear components of temperature. Based on the temperature map, which was normalized by applying a regression model, the difference in air temperature between the measured and normalized data was calculated:

$$T_{a \cdot t} = T_{m \cdot t} - T_{n \cdot t} \tag{3}$$

$$AT_{a \cdot t} = AT_{a \cdot (t-8)} + i \times T_{a \cdot t} \tag{4}$$

where $T_{a \cdot t}$ is the air temperature anomaly, and $T_{m \cdot t}$ and $T_{n \cdot t}$ are the measured air temperature and normalized air temperature, respectively. Normalized air temperature means the temperature that can reflect changes of the environmental conditions (altitude, longitude, latitude, etc.) on the natural land surface (forest, agricultural land, etc.). $AT_{a \cdot t}$ is the accumulated value of air temperature anomalies from DoY1 until DoYt. i is the DoY interval coefficient, which ranges from 1 to 8.

The EVI was calculated from the MODIS surface reflectance product based on Equation (5):

$$EVI = G \times \frac{\rho NIR - \rho RED}{\rho NIR + C1 \times \rho RED - C2 \times \rho BLUE + L} \tag{5}$$

where $\rho NIR$, $\rho RED$, and $\rho BLUE$ are near the infrared, red, and blue bands in MOD09A1, respectively. $L$ is the canopy background adjustment, $C1$ and $C2$ are the coefficients, and $G$ is the gain factor. Then, a bit-pattern analysis was implemented to remove the cloudy pixels identified by the information in the state flags (16-bit unsigned integer). In order to deduct green-up dates from the EVI, we used a sigmoid-based equation [63]. In the sigmoid model, phenological transition dates were attained by obtaining maximum value in the rate of change of curvature.

### 2.5. Statistical Analysis

We measured the linear regression coefficient and Pearson's correlation coefficient to deduct relationship between air temperature and first spring phenological events. In addition, we used one-way analysis of variance (ANOVA) followed by Tukey's HDC post hoc test to compare the difference of mean air temperature anomaly among the three land-use types. The linear regression analysis was performed by using Microsoft Excel 2016 software (Microsoft Corp., Redmond, WA, USA), and Pearson's correlation test and ANOVA test were performed by using SigmaPlot 12.0 software (Systat Software Inc., Chicago, IL, USA).

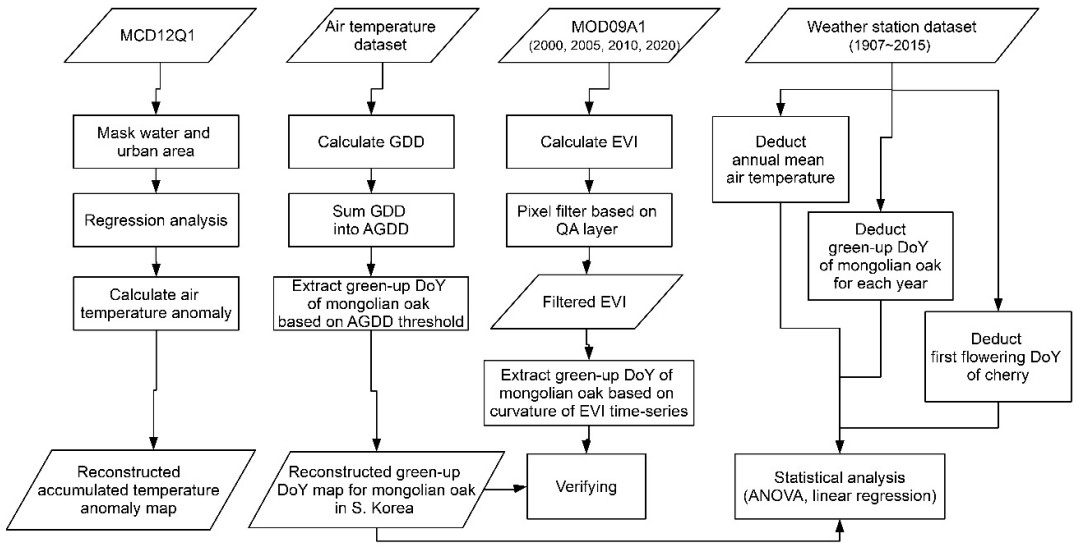

**Figure 2.** Flowchart showing the overall process of data processing and experiment.

## 3. Results

### 3.1. Spatial Distribution of Isopleth of DoY for AGDD 159

Spatial distribution of isopleth of DoY for AGDD 159 was expressed on the topography and land-use type maps (Figure 3). DoY for AGDD 159 tended to be earlier in the sites with lower latitude

and altitude, where area located on the southern and western parts of South Korea. On the other hand, an effect of urbanization on advancement of green-up dates was also confirmed from several verification sites such as 3, 4, 11, 12, 19, 23, 24, and so on (Figure 3).

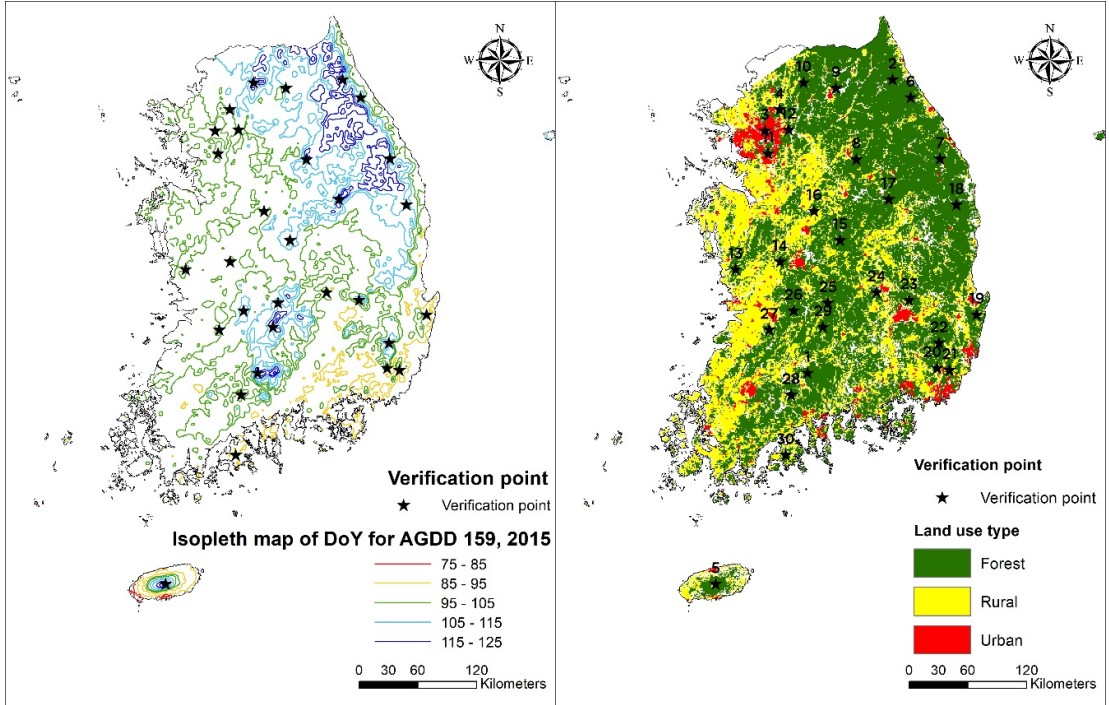

**Figure 3.** The isopleth map of day of year (DoY) for accumulated growing degree days (AGDD) 159 overlapped on the topography (left) and land-use type (right) maps.

### 3.2. Relationship Between Air Temperature and Phenological Events

We compared green-up dates of Mongolian oak and air temperature among major Mongolian oak forests in each season (winter, DoY 1~59; spring, DoY 60~151; total, DoY 1~184) throughout South Korea to investigate relationship between unfolding start date of Mongolian oak and air temperature. We deducted mean air temperature and green-up date (DoY_AGDD) from the MODIS image as the meteorological stations are located far from Mongolian forests.

Figure 4 shows the relationship between MODIS-derived air temperature and the green-up start date for each sampling site of Mongolian oak forests (N = 30; Figure 1). At the seasonal scale, the date had the highest correlation with mean air temperature in spring ($R^2$ = 0.87), followed by the whole season ($R^2$ = 0.84), the winter season ($R^2$ = 0.59). The result indicated that if the mean air temperature in the spring season rises by 1 °C, Mongolian oak will leaf out earlier by 3.84 DoY. Similarly, mean air temperature during DoY 1~184 was highly correlated with the date ($R^2$ = 0.84), and the regression coefficient indicated that the date will move back by 3.58 DoY in accordance with the rising of mean air temperature by 1 °C during DoY 1~184.

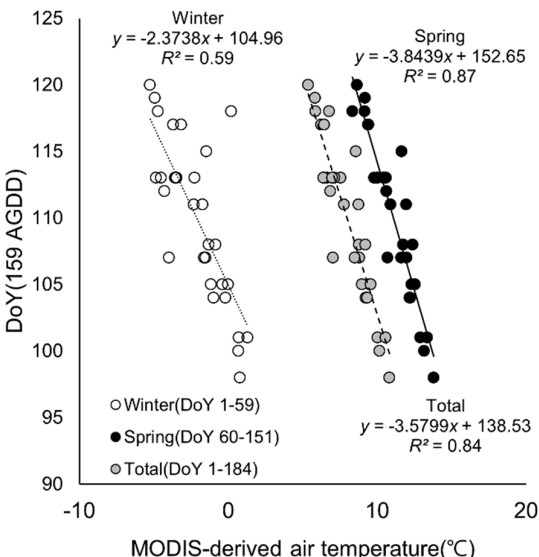

**Figure 4.** The relationship between mean moderate resolution imaging spectroradiometer (MODIS)-derived air temperature (winter: January to February, spring: March to May, total: January to June) and green-up DoY$_{AGDD}$ for 30 sampling sites of Mongolian oak forests in South Korea.

Green-up start dates from MODIS-derived AGDD in 2015 and EVI in 2000, 2005, 2010, and 2015 were closely related with each other (Figure 5). Green-up start dates from MODIS-derived EVI in 2000, 2005, 2010, and 2015 were also closely related with each other (Figure 6).

Another comparison was carried out to establish the relationship between the seasonal mean air temperature measured from meteorological stations (N = 93; Figure 1) and the green-up DoY determined by the AGDD threshold derived from measured air temperature. The result indicated that the seasonal mean air temperatures were significantly related to the green-up DoY ($p \leq 0.01$) (Figure 7). Among the results compared, the coefficient of determination was the highest in the winter season ($R^2 = 0.78$), followed by the whole season ($R^2 = 0.75$), and the spring season ($R^2 = 0.65$).

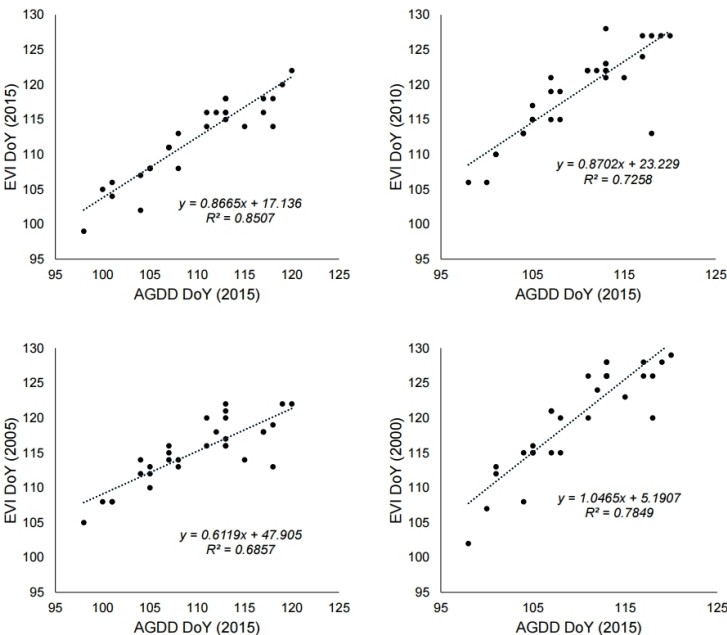

**Figure 5.** The relationship between the AGDD DoY and the enhanced vegetation index (EVI) DoY for 30 sampling sites of Mongolian oak forests in South Korea. Green-up DoY deducted based on AGDD

in 2015 showed significant correlation with green-up DoY deducted based on EVI in 2000, 2005, 2010, and 2015.

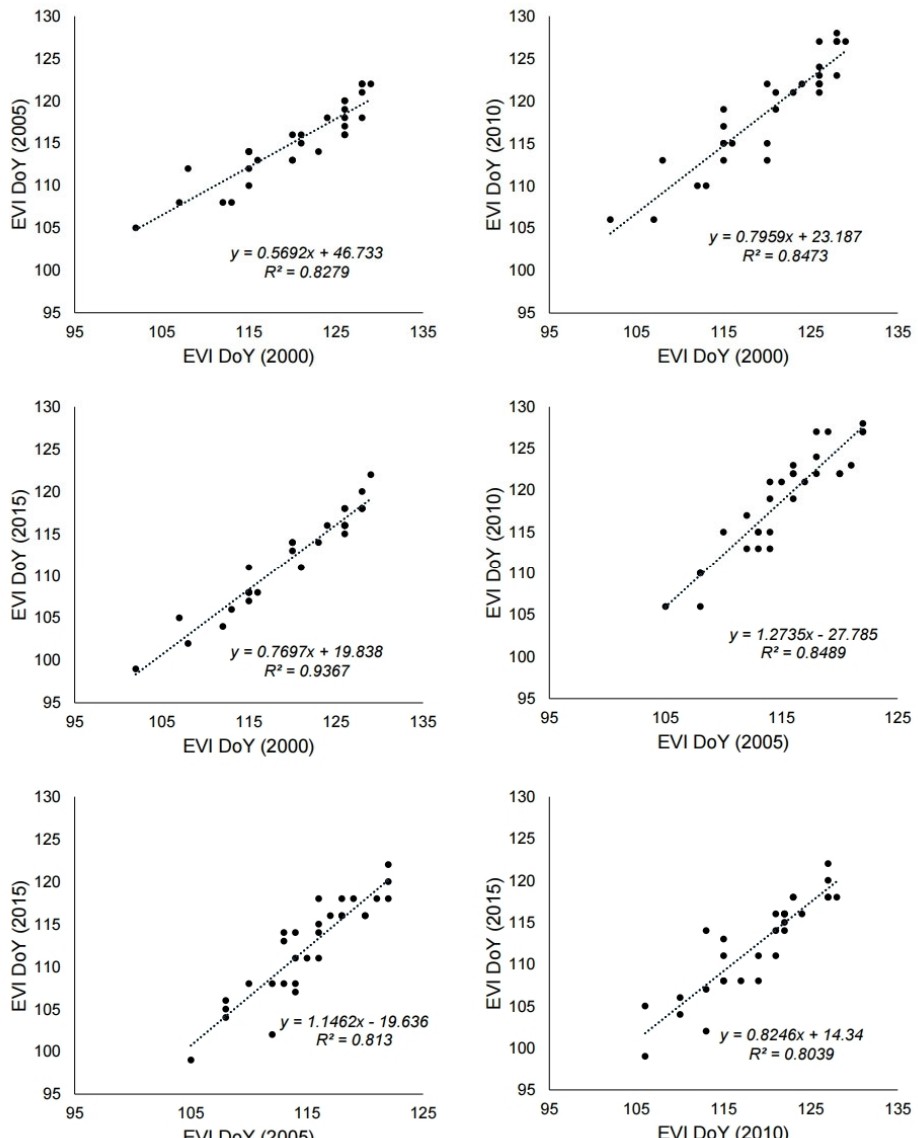

**Figure 6.** The relationship between green-up DoY deducted based on EVI for 30 sampling sites of Mongolian oak forests in 2000, 2005, 2010, and 2015 in South Korea. Green-up DoY in each year showed significant correlation with each other.

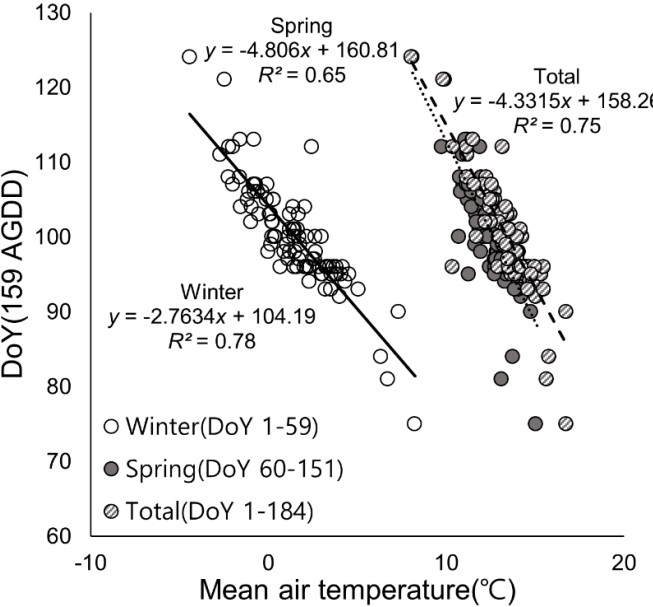

**Figure 7.** The relationship between the seasonal mean air temperatures measured from 93 meteorological stations and the green-up DoY determined by the AGDD threshold.

## 3.3. Change of Green-Up Date during the Past Century

To determine the relationship between vegetation phenology and air temperature on a longer temporal scale, we analyzed the response of vegetation phenology due to the rise of the mean air temperature during the past century. The analysis was carried out for five meteorological stations that had recorded the DoY of the phenological event (first flowering DoY of cherry; Figure 8) and the air temperature during the period (Figure 9). The flowering date of cherry (Figure 8) became earlier and air temperature (Figure 9) was risen significantly. Consequently, both factors showed a negative correlation (Table 2). Leaf green-up date of Mongolian oak based on the AGDD threshold showed the same trend as the flowering date of cherry (Figure 10, Table 2). Therefore, the green-up DoY$_{AGDD}$ of Mongolian oak was highly correlated to both the annual mean air temperature and the first flowering date of cherry as a phenological event ($p \leq 0.001$; Table 3).

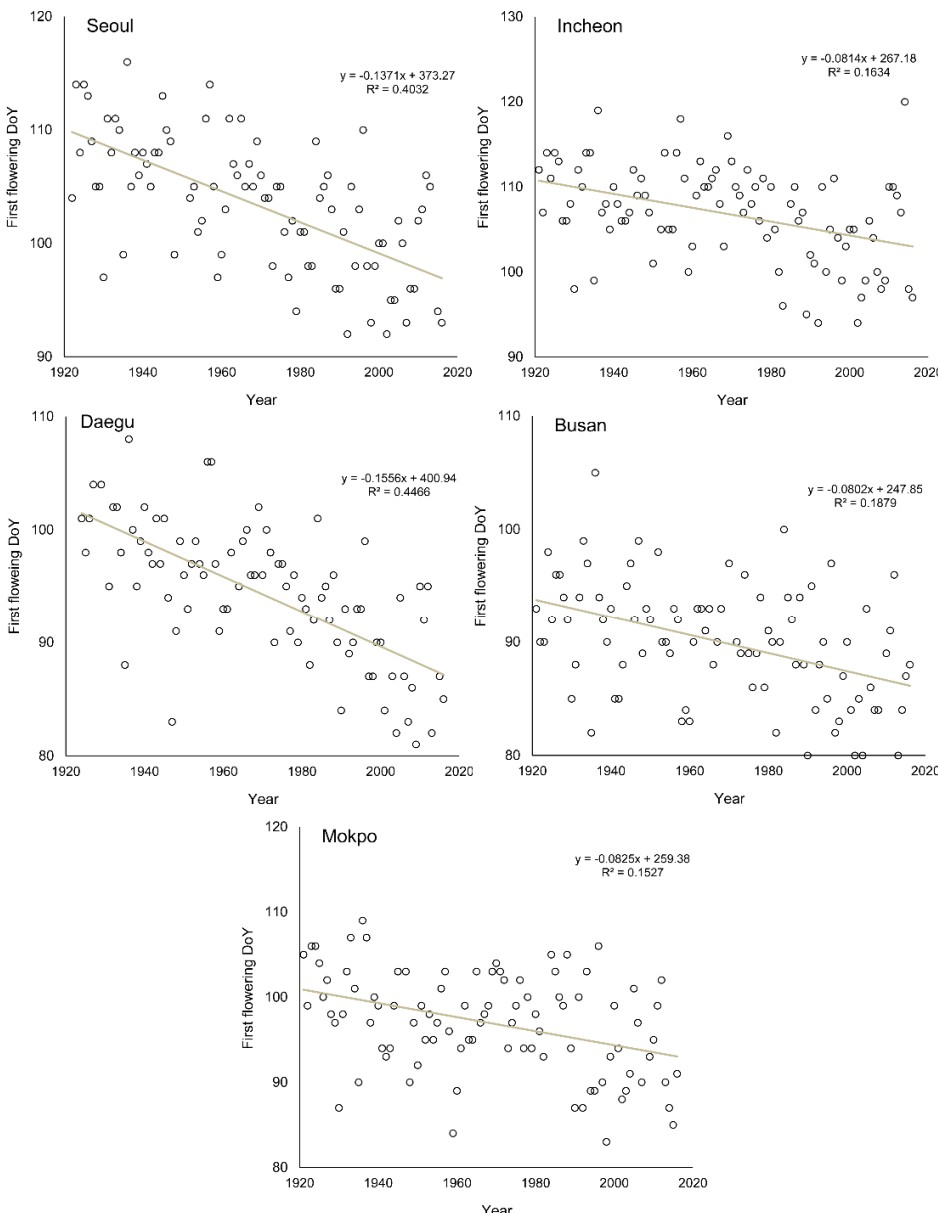

**Figure 8.** Changes of the first flowering DoY of cherry during the past century in major cities of South Korea.

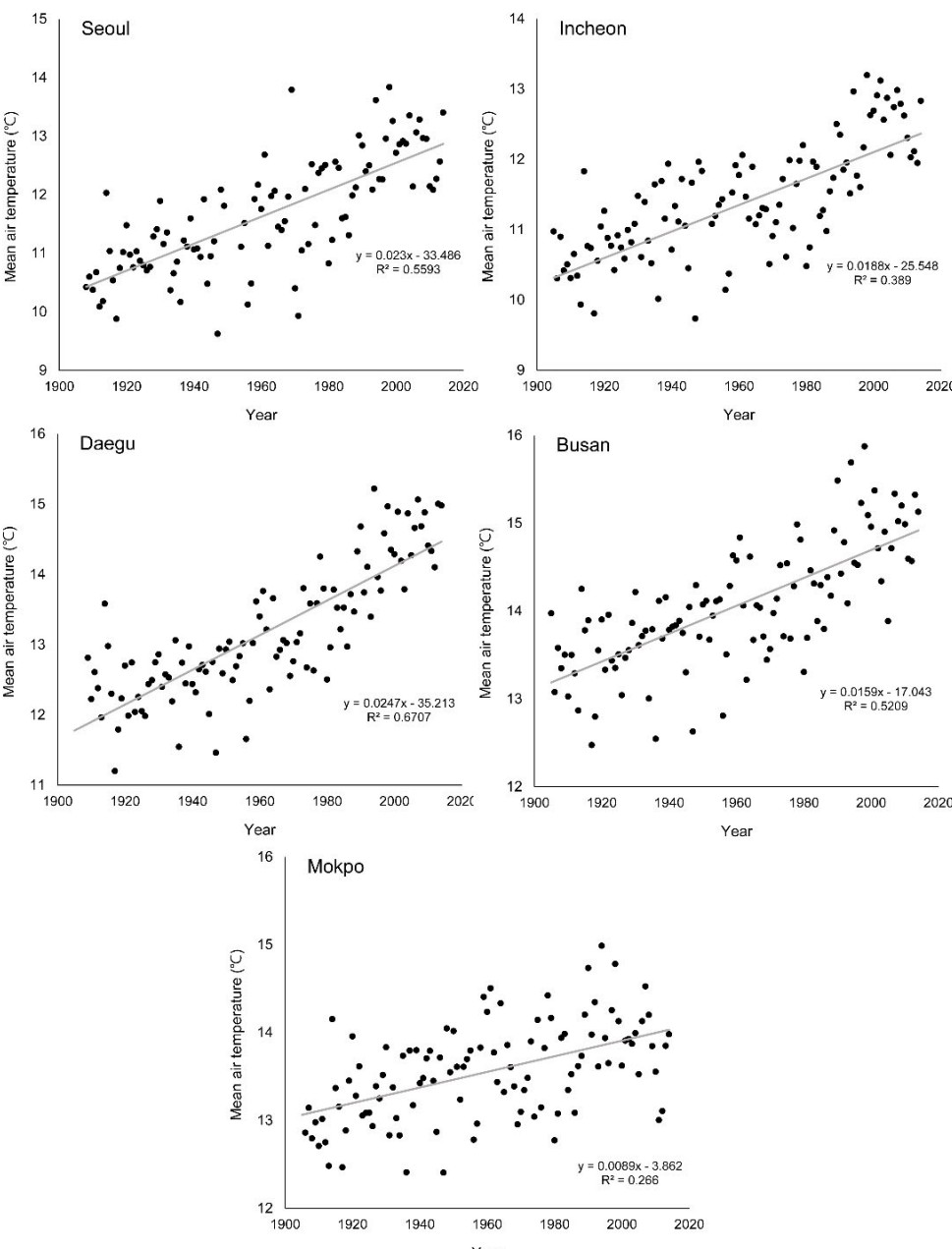

**Figure 9.** Changes of the annual mean air temperature during the past century in major cities of South Korea.

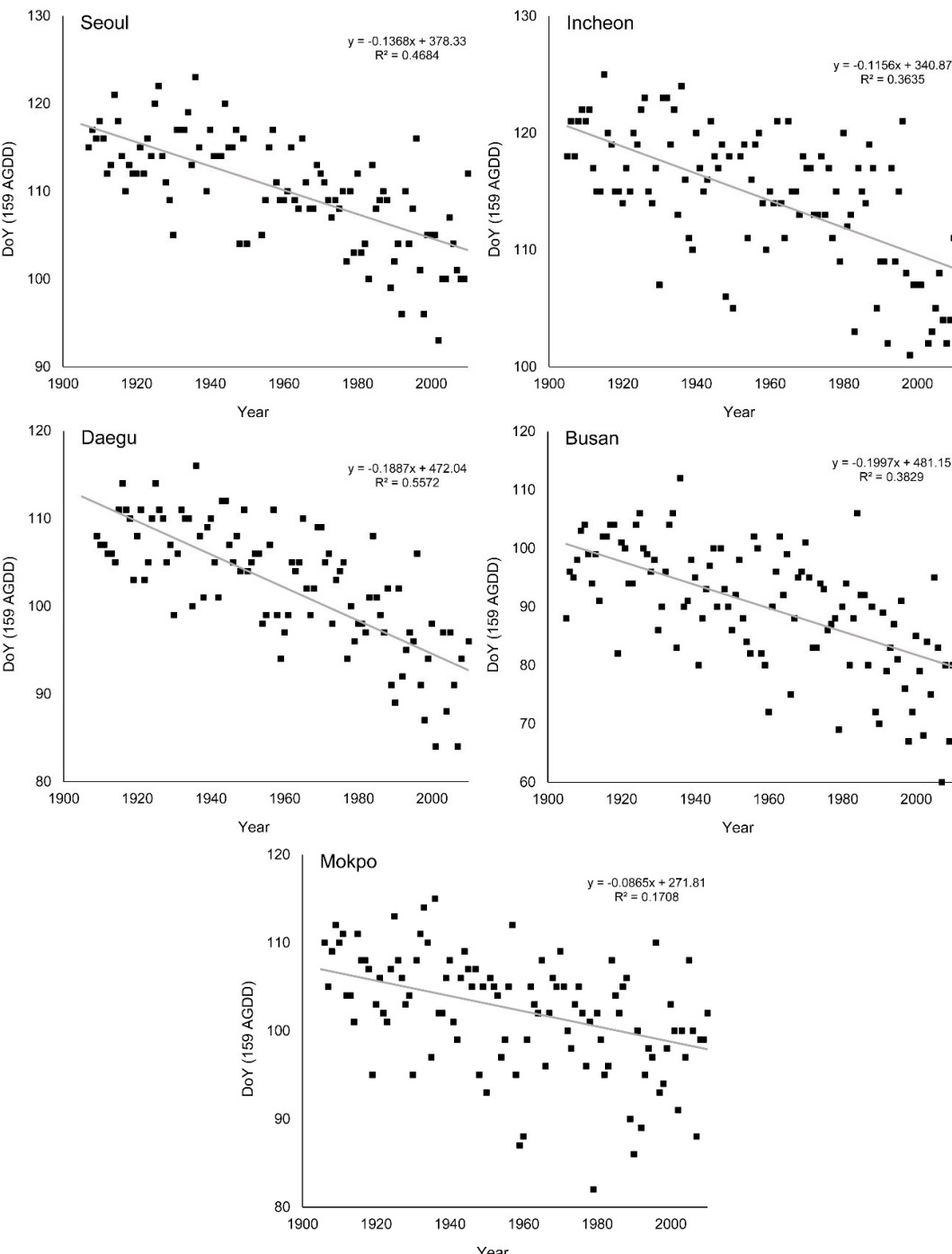

**Figure 10.** Changes of the simulated green-up dates of Mongolian oak, which require 159 AGDD during the past century using the air temperature measured in the meteorological stations in major cities of South Korea.

**Table 2.** The rate of change for the annual mean air temperature and the phenological event dates during the past century.

| City | Linear Regression Coefficient (/100 Years) | | |
| --- | --- | --- | --- |
| | Annual Mean Air Temperature ($R^2$) | DoY$_{AGDD}$ ($R^2$) | First flowering DoY of Cherry ($R^2$) |
| Seoul | 2.3 (0.56 **) | −13.68 (0.47 **) | −13.71 (0.40 **) |
| Incheon | 1.8 (0.39 *) | −11.56 (0.36 *) | −8.10 (0.16 *) |
| Daegu | 2.5 (0.67 **) | −18.87 (0.56 **) | −15.56 (0.45 **) |
| Busan | 1.6 (0.52 **) | −19.97 (0.38 *) | −8.00 (0.19 *) |
| Mokpo | 0.9 (0.27 *) | −8.65 (0.17 *) | −8.30 (0.15 *) |

* Significant at $p \leq 0.05$; ** significant at $p \leq 0.01$.

**Table 3.** The relationship between green-up DoY$_{AGDD}$ of Mongolian oak and the annual mean air temperature and the first flowering DoY of cherry.

| City | N | Coefficient of Pearson's Correlation | |
| --- | --- | --- | --- |
| | | Annual Mean Air Temperature (°C) | First Flowering DoY of Cherry |
| Seoul | 87 | −0.754 *** | 0.913 *** |
| Incheon | 92 | −0.574 *** | 0.731 *** |
| Daegu | 87 | −0.832 *** | 0.854 *** |
| Busan | 91 | −0.789 *** | 0.838 *** |
| Mokpo | 91 | −0.716 *** | 0.756 *** |
| Total | 299 | −0.823 *** | 0.913 *** |

*** Significant at $p \leq 0.001$.

*3.4. Temperature Anomaly due to Topography and Land-Use Intensity*

Figure 11 shows the mean air temperature anomaly in each season. From the winter to the spring, the area that had high temperature anomaly was mostly distributed in the east coast region of South Korea. It is because this region experiences a temperature gradient due to the mountain ranges with high altitude, which block off the northwesterly wind from the Siberian anticyclone during the period. This phenomenon was also seen in the southern part of Jeju Island, the southernmost island of South Korea, which has a high mountain (Mt. Halla) long-stretched from east to west in the center.

Aside from the areas that were affected by seasonal wind, there were many hotspots distributed as a point pattern (Figure 11). The reason that the air temperature was higher than the surrounding area in these areas is due to the urban heat island (UHI) effect from the intensive land-use by human beings.

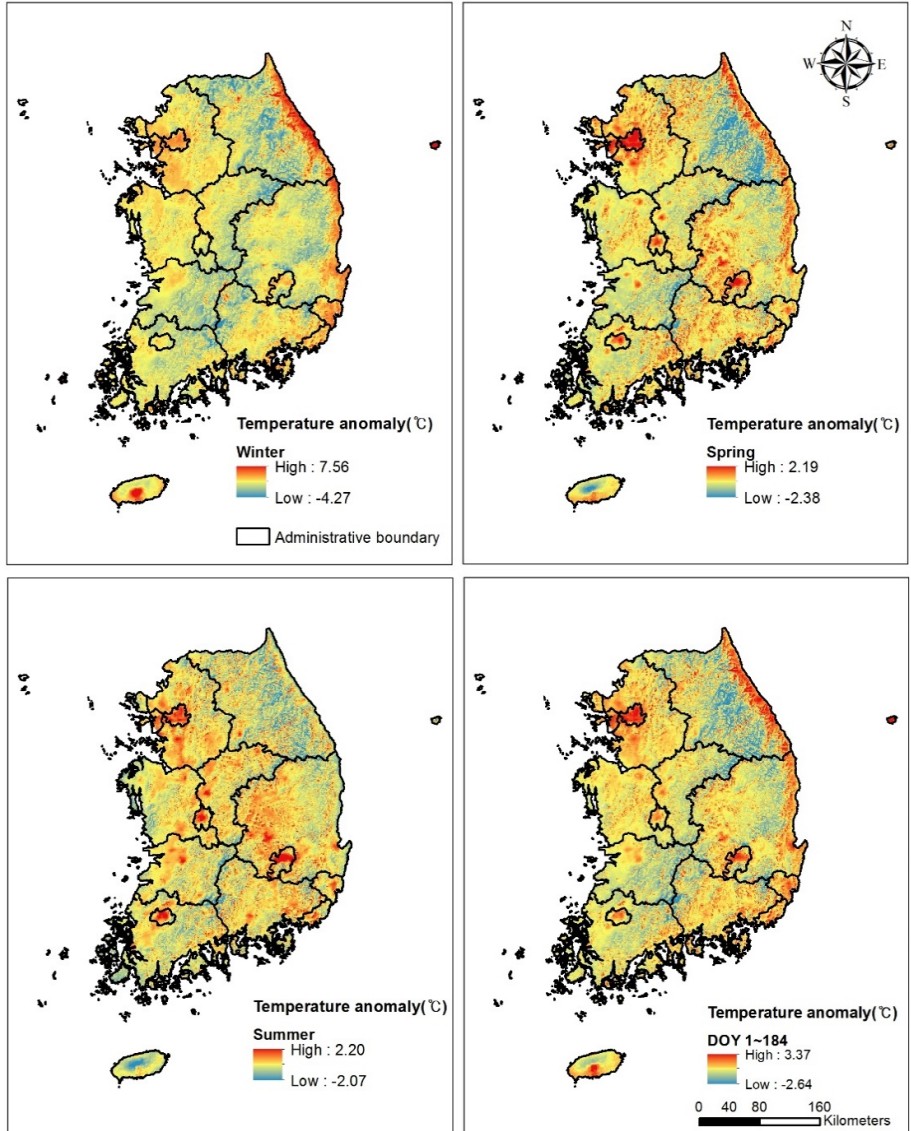

**Figure 11.** The spatial distribution of air temperature anomalies during winter (**upper left**), spring (**upper right**), summer (**lower left**), and DoY 1~184, 2015 in South Korea (**lower right**).

The results of an analysis of variance (one-way ANOVA) and a post hoc analysis (Tukey honestly significant difference; Tukey HSD) indicated that there was a difference (F = 1889.62) between groups, urban, rural, and forest areas at the significance level of 0.001 (Table 4), and the difference was clear ($p \leq 0.001$) (Table 5). In particular, the urban area appeared to have the higher difference compared to the other groups.

Figure 12a shows the time-series patterns of the mean air temperature anomaly for each land-use type. It can be seen that 8-d anomalies in urban areas were always higher than 0 °C. In contrast, anomalies in the other land-use types fluctuated around approximately 0 °C in a time-series. Consequently, the temperature anomaly in urban area consistently increased differently from that in areas that belonged to the other land-use type (Figure 12b). Hence, the time to reach the AGDD threshold in the urban area was much shorter than the time in the other areas.

**Table 4.** The one-way ANOVA table shows the difference of mean air temperature anomaly among the three land-use types (urban, rural, forest).

|  | Sum of Squares | Df | Mean Square | F | Sig. |
|---|---|---|---|---|---|
| Between Groups | 1049.47 | 2 | 524.73 | 1889.62 | 0.000 *** |
| Within Groups | 23,910.99 | 86,106 | 0.28 |  |  |
| Total | 24,960.46 | 86,108 |  |  |  |

*** Significant at $p \leq 0.001$.

**Table 5.** The results of Tukey's HSD(honestly significant difference) post hoc test. Std. error: Standard error, Sig.: Significance value.

| CLASS | CLASS | Mean Difference | Std. Error | Sig. | 95% Confidence Interval | |
|---|---|---|---|---|---|---|
|  |  |  |  |  | Lower Bound | Upper Bound |
| Forest | Rural | 0.06 * | 0.004 | 0.000 *** | 0.0506 | 0.0689 |
|  | Urban | −0.46 * | 0.008 | 0.000 *** | −0.4802 | −0.4419 |
| Rural | Forest | −0.06 * | 0.004 | 0.000 *** | −0.0689 | −0.0506 |
|  | Urban | −0.52 * | 0.008 | 0.000 *** | −0.5407 | −0.5009 |
| Urban | Forest | 0.46 * | 0.008 | 0.000 *** | 0.4419 | 0.4802 |
|  | Rural | 0.52 * | 0.008 | 0.000 *** | 0.5009 | 0.5407 |

* Significant at $p \leq 0.05$; *** significant at $p \leq 0.001$.

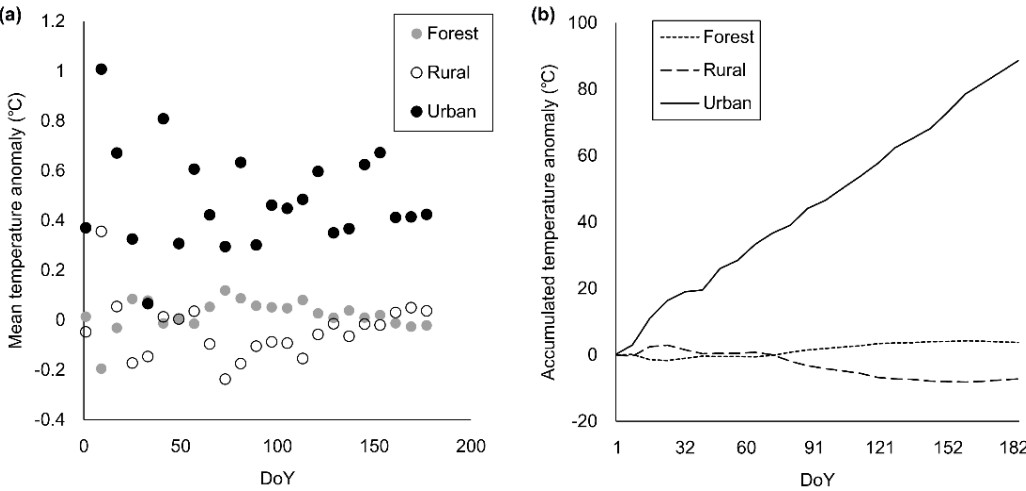

**Figure 12.** The time-series patterns of the 8-d mean air temperature anomaly (**a**) and accumulated air temperature anomaly (**b**) for land-use types.

## 4. Discussion

### 4.1. Utility of MODIS Images as a Tool for Phenology Research

Lim et al. [60,63] confirmed the leaf unfolding start date of Mongolian oak from the digital cameras installed in Mongolian oak community of five locations with different ecological conditions and reported that these results confirmed at the field were consistent with the leaf unfolding start date based on AGDD derived from the time series analysis of the satellite image (MODIS) temperature data.

Air temperatures obtained through satellite image analysis for 30 Mongolian oak forests selected throughout the whole national territory in this study reflected the temperature changes occurring from the difference in latitude and altitude of each site well. Furthermore, the leaf unfolding start dates of the Mongolian oak based on both AGDD (Figure 4) and EVI in each site (Figure 5) were closely correlated with each other. The leaf unfolding start dates of the Mongolian oak deduced based on EVI in different years, 2000, 2005, 2010, and 2015, were also closely correlated with each other (Figure 6). In addition, the result showed a similar trend to the relationship between the mean temperature and the green-up DoY determined by the AGDD threshold derived from the air temperature measured by 93 weather stations (Figure 7).

As the result of analysis on the yearly changes of the unfolding start date of Mongolian oak was based on AGDD calculated from air temperature data for about 100 years measured by the Korea Meteorological Administration, the date was assessed to have advanced by 9–20 days on a national basis (Table 2, Figure 10). These results have been inversely proportional to changes in the average temperature measured by the Korea Meteorological Administration over the past 100 years, and positively correlated with the cherry blossom date measured by the Korea Meteorological Administration (Table 3, Figures 8 and 9).

As was shown above, the leaf unfolding start dates of Mongolian oak derived from satellite image analysis were consistent with the results identified visibly in the field sites [60], and well reflected the spatial temperature variation. Furthermore, the dates also tended to be proportional to the long-term annual mean temperature and the change in the cherry blossom date measured directly. In this regard, the phenology study through MODIS image analysis was evaluated to have methodological validity.

The relationship between air temperature or degree days and phenophases, especially flowering and leaf unfolding, is well known and has been widely reviewed [5,60]. As temperature has been known as a major driver of phenology [25,69], AGDD has a long history of use in predicting plant and insect phenology in agriculture [60,70,71].

Traditionally, the estimation of air temperature has depended on ground measurements at point levels such as meteorological stations and ground surveys. The ground-measured air temperatures were spatially interpolated using a GIS with conventional spatial statistics techniques such as kriging, inverse distance weighting (IDW), spline, and so on for expanding the estimation to the polygon level. Although the development of GIS and spatial statistics have led to a drastic refinement in the interpolated result, there is a severe weakness due to the limited number of the points [60,72].

But remote sensing is an alternative data source since remotely-sensed imagery is intrinsically spatialized [59]. In particular, MODIS provides an abundant series of land surface temperature (LST) products with different spatial and temporal resolutions from both terra and aqua platforms [57]. Previous studies have shown that the LST data measured by MODIS can be successfully used for linear regression estimates of air temperatures at a regional scale [60,73–75].

### 4.2. Climate Change and Vegetation Phenology

Based on the records at major cities of South Korea, mean air temperature has risen between 0.9 to 2.5 °C (on average 1.8 ± 0.6 °C) during the past century (Figure 8). During the period, cherry has flowered earlier, from 8.00 to 15.56 days (on average 10.7 ± 3.6 days) (Figure 8, Table 2). Applying the green-up DoY determined by the AGDD threshold, Mongolian oak has leafed out earlier, from 8.65 to 19.97 days (on average 14.5 ± 4.3 days) (Figure 10, Table 2).

Changes in climate conditions influence the energy and time budgets of individual organisms and thus can directly alter the timing of such events. Therefore, phenology is utilized as a valuable tool for diagnosing the biological impacts of climate change [8,18]. Climate changes may also influence phenology indirectly, as organisms use environmental cues, such as temperature or rainfall, to regulate the timing of specific events within their annual cycle [76]. Changes in phenology have emerged as one of the most conspicuous responses of biota to recent climatic warming [43,77–79]. Many long-term phenological data sets provide persuasive evidence of significant temporal advances in several phenological events across a wide range of biological species from various regions [80,81]. The consistency of this pattern is shown by the phenological advances found in flowering and leaf unfolding times [82–86]. In general, the most remarkable phenomena have been found in the phenophases occurring in the spring season [43].

Many data usually demonstrate that these changes are mainly a product of temperature increase rather than of other aspects of the weather [6,8,87–94]. However, precipitation (Mediterranean vegetation) [85,95], the timing of snowmelt and the temperatures that follow snowmelt (high-latitude and high-altitude ecosystems), the amount and seasonal variability of precipitation, the duration of the dry season, solar radiation (tropical forests) [96–102], and photoperiod and winter chilling requirements (some temperate tree species) [103–107] are also known as critical factors that regulate spring phenology.

*4.3. Urban Heat Island Effect and Vegetation Phenology*

The results of this study show that urbanization affected not only temperature rise (Figure 3) but also phenology Figures 8 and 9, Table 2). Green-up date of Mongolian oak obtained from five major cities, 93 meteorological stations, and 30 natural forests was earlier in the mentioned order (Figures 8 and 9, Table 2) and the order was attributed to the degree of urbanization of the area that the data was collected. The results from this study agreed with the results from numerous previous studies [23,51,108,109]. The results showed that urban areas experienced a greater accumulated air temperature than the rural and forest areas in similar geographical conditions, such as altitude, latitude, and longitude (Figure 12b). Consequentially, the experience caused phenological events in urban area to occur earlier than in the other areas. For example, in 2015, the green-up date of Mongolian oak forests on Mt. Nam, located in the center of Seoul, the capital of South Korea, was DoY 98 [60], which was earlier by approximately 10 days compared to the date in the surrounding rural and forest areas (Figure 4). Such an earlier spring green-up was because Mongolian oak on Mt. Nam had experienced the accumulated air temperature anomaly of 188.26 (Figure 12).

Urbanization is considered as a main driver of climate change [110]. Land transformation and increasing impervious surface cover increase local temperatures and alter ecosystem processes such as the carbon cycle [111–115]. Built surfaces generally absorb more solar radiation than vegetation; they are also impervious, covering the soil and preventing heat dissipation from evapotranspiration [116,117]. Thus, land use pattern can account for much of the temperature variation [28,118–121]. The urban energy budget also controls temperature variance significantly [122–124].

Urbanized areas are also exposed to climate change from greenhouse gas-induced radiative forcing, and localized effects from urbanization such as the urban heat island. Warming and extreme heat events due to urbanization and increased energy consumption are simulated to be as large as the impact of doubled $CO_2$ in some regions [41]. Urban micro-climates have long been recognized [125]. Observational evidence showed trends in urban heat islands in some locations of a similar magnitude or greater than that from greenhouse gas-forced climate change [126,127]. In the phenological study, urban areas represent important study fields because their warmer conditions allow for an assessment of the future potential impacts of climate change on plant development [128] UHI-induced increases in temperature can affect vegetation phenology both within and around cities [49]. Lee et al. [129] and Jung et al. [18] reported that abnormal shoot growth appeared more frequently, and shoot length was longer, in the hotter urban center than in the urban fringe or the suburban greenbelt and the frequency of abnormal shoots and their lengths were closely correlated with the urbanized ratio (positively) and with the vegetation cover of land expressed as NDVI

(Normalized difference vegetation index; negatively) in Seoul. A study carried out by Zipper et al. [51] showed that the length of the urban growing season in Madison, Wisconsin of the USA is approximately five days longer than in the surrounding rural areas, and the UHI impacts on growing season length are relatively consistent from year to year. These studies have indicated that vegetation phenology is relevant to the UHI effect and the potential impacts of climate change on urban climate [130].

## 5. Conclusions

The changes in phenology due to climate change vary depending on time, space, degree of human intervention, and so on. Most studies for plant phenology have been carried out focusing on the specific factors. However, various factors need to be considered in order to understand plant phenology at certain times. In this study, we tried to clarify the changes in phenology of the Mongolian oak in various aspects by utilizing satellite image data. It is necessary to study long-term trends to understand the relationship between climate change and phenology, but the method by which satellite images are utilized inherently includes the fundamental problem that the period of analysis is short. These problems can be supplemented by taking into account environmental factors such as temperatures, which play an important role in phenological events of plant, rather than using the specific reflective properties of the plant. Information on temperatures measured and the blooming dates of cherry blossoms observed for more than 100 years in Korea can be used as very powerful sources for studying the changes of phenology due to climate change. In this study, we clarified the relationship between air temperature and spring green-up date of Mongolian oak by applying a meteorological indicator (AGDD) derived from MODID LST imagery in 30 sampling sites of the Mongolian forest throughout the whole national territory of South Korea. Furthermore, we clarified a correlation between both factors by applying AGDD based on the records of 93 meteorological stations. In addition, the results of this study were confirmed by the data that flowering date of cherry became earlier and responded negatively on the rise of mean air temperature during the past century based on the records at five meteorological stations. Through this study we showed that the study for phenology can be carried out through meteorological indicators derived from satellite images. We could also confirm that long-term trends in the changes could be deduced by utilizing the additional observation data. Furthermore, the results obtained from this study showed that the spatiotemporal changes in environmental conditions cause obvious changes in plant phenology. In addition, the result of this study showed that change of microclimate due to the intense land use in the urban area also caused change of vegetation phenology. Overall, we extended the spatiotemporal scales of the phenological study by applying the remote sensing image interpretation and the spatial statistics techniques. We expect our findings could be used to predict long-term changes in ecosystems due to climate change.

**Author Contributions:** Conceptualization, C.H.L. and C.S.L.; methodology, C.H.L. and C.S.L.; software, C.H.L. and N.S.K.; validation, S.H.J. and C.S.L.; formal analysis, C.H.L. and A.R.K.; investigation, C.H.L. and A.R.K.; resources, N.S.K. and C.S.L.; data curation, C.S.L. and S.H.J.; writing—original draft preparation, C.S.L.; writing—review and editing, C.S.L.; visualization, C.H.L.; supervision, C.S.L.; project administration, C.H.L. All authors have read and agreed to the published version of the manuscript.

**Funding:** This research received no external funding.

**Conflicts of Interest:** The authors declare no conflict of interest.

## Abbreviations

| | |
|---|---|
| GDD | Growing degree days set as 5 °C |
| AGDD | Accumulated growing degree days |
| EVI | Enhanced vegetation index (EVI) |
| MODIS LST | Moderate resolution imaging spectroradiometer land surface temperature |
| DoY$_{AGDD}$ | Day of year needed to reach the AGDD threshold (159 °C·d) |
| DoY (159 AGDD) | Day of year required to reach AGDD 159 °C necessary for the leaf unfolding |
| AGDD 159 °C | Accumulated growing degree days 159 °C necessary for the leaf unfolding |

## Appendix A

**Table A1.** The DoY reached the AGDD 159 and the green-up DoY in 2015 for 30 sampling sites of *Quercus mongolica* forest throughout South Korea. Max. K: DoY required for green-up start based on curvature K, Diff.: Difference between days deducted based on AGDD 159 and curvature K.

| No. | Site No. | Lon. | Lat. | Elevation (m) | Aspect (°) | Max. K | DoY for AGDD 159 | Diff. |
|---|---|---|---|---|---|---|---|---|
| 1 | Mt. Jiri | 128.46 | 38.04 | 985 | 90 | 117 | 121 | 4 |
| 2 | Mt. Jeombong | 127.42 | 38.01 | 906 | 78 | 118 | 123 | 5 |
| 3 | Mt. Nam | 127.80 | 37.95 | 141 | 13 | 98 | 98 | 0 |
| 4 | Gwangneung | 128.67 | 37.87 | 230 | 42 | 104 | 107 | 3 |
| 5 | Mt. Halla | 127.16 | 37.75 | 1315 | 317 | 118 | 125 | 7 |
| 6 | Mt. Odae | 127.26 | 37.57 | 690 | 148 | 113 | 116 | 3 |
| 7 | Mt. Deokwang | 126.99 | 37.55 | 878 | 331 | 113 | 119 | 6 |
| 8 | Mt. Chiak | 127.03 | 37.34 | 1090 | 298 | 119 | 123 | 4 |
| 9 | Mt. Majeok | 129.01 | 37.31 | 513 | 2 | 111 | 109 | −2 |
| 10 | Gookmangbong | 128.05 | 37.3 | 1124 | 240 | 113 | 119 | 6 |
| 11 | Mt. Gwanggyo | 128.43 | 36.94 | 429 | 21 | 107 | 108 | 1 |
| 12 | Mt. Yebong | 129.20 | 36.89 | 423 | 54 | 108 | 109 | 1 |
| 13 | Mt. Ami | 127.57 | 36.82 | 306 | 198 | 104 | 107 | 3 |
| 14 | Mt. Gyeryong | 127.88 | 36.56 | 527 | 304 | 107 | 109 | 2 |
| 15 | Mt. Sokri | 127.20 | 36.35 | 907 | 67 | 113 | 115 | 2 |
| 16 | Mt. Doota | 126.69 | 36.27 | 408 | 164 | 105 | 106 | 1 |
| 17 | Mt. Sobaek | 128.29 | 36.08 | 907 | 75 | 120 | 126 | 6 |
| 18 | Mt. Donggo | 128.66 | 36.01 | 900 | 187 | 112 | 116 | 4 |
| 19 | Mt. Moojang | 127.75 | 35.98 | 320 | 90 | 100 | 97 | −3 |
| 20 | Mt. Maebong | 127.36 | 35.9 | 607 | 246 | 105 | 103 | −2 |
| 21 | Mt. Cheongsong | 129.43 | 35.88 | 523 | 225 | 101 | 97 | −4 |
| 22 | Mt. Gaji | 127.69 | 35.76 | 1126 | 316 | 113 | 116 | 3 |
| 23 | Mt. Palgong | 127.09 | 35.72 | 860 | 186 | 107 | 114 | 7 |
| 24 | Mt. Geumoe | 129.01 | 35.62 | 709 | 111 | 108 | 105 | −3 |
| 25 | Mt. Baekwoon | 128.98 | 35.39 | 895 | 203 | 113 | 116 | 3 |
| 26 | Mt. Woonjang | 129.12 | 35.37 | 969 | 235 | 111 | 114 | 3 |

| 27 | Mt. Moak | 127.53 | 35.33 | 721 | 242 | 115 | 109 | −6 |
| 28 | Mt. Birae | 127.35 | 35.13 | 571 | 320 | 105 | 103 | −2 |
| 29 | Mt. Namdeokyou | 127.30 | 34.58 | 1197 | 181 | 117 | 121 | 4 |
| 30 | Mt. Jogye | 126.55 | 33.38 | 266 | 117 | 101 | 97 | −4 |

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
