# Peer review of "Monitoring for Changes in Spring Phenology at Both Temporal and Spatial Scales Based on MODIS LST Data in South Korea"

_remotesensing, doi:10.3390/rs12203282_

Round 1

Reviewer 1 Report

  1. Why Mongolian oak was chosen as the research plant? What are the advantages of Mongolian oak compared with other plants? Have you compared the results of other vegetations? It is suggested to add a description of these contents, which can be put into the discussion section.
  2. Lines 110-111, why use "cherry blossom data" to verify "the change of green-up dates of Mongolian oak over year deducted from temperature variation"? It looks like two separate experiments. Why can we verify the correctness of the conclusion that Mongolian oak's green-up date is advanced by cherry blossom data? What is the relationship between the phenological phases of the two plants? What is the rationality of this verification method?
  3. The method part of the paper is too brief and the structure is chaotic. In section 2.2 Experimental design, the research data used was mentioned many times during the description of the experiment, but the description was not accurate enough to allow people to clearly know what data was used. And the calculation methods and processes of GDD, AGDD, AGDD threshold and green up dates should be introduced, even if the previous research methods are used, they should also be described in this paper. At the same time, it is suggested to add data processing diagram and experimental flow chart to make the research structure clearer.
  4. Lines 139-141 refer to the 8-d interval air temperature map dataset. This data quotes the results of previous research. It is not described in detail here, including time resolution, spatial resolution, accuracy, etc. It is recommended to supplement the data description.
  5. Is there some repeats between the parts of section 2? It is suggested that when the experimental process and data are explained in detail, the repeated contents in each part should be removed, and the key points of each part should be highlighted. At the same time, the connection between the steps should be enhanced, which is helpful to strengthen the context connectivity.
  6. English abbreviations appear in many places in the paper, such as "AGDD", "DoYAGDD", "DoY (159 AGDD)", "AGDD 159°C", etc. It is recommended to supplement the comparison table of proper noun abbreviations, and explain their representatives when they first appear The meaning and purpose of the research to enhance the reading experience of readers.
  7. There are some mistakes in the text, please check. Lines 255-256, "Flowering date of cherry (Figure 5) became earlier and air temperature (Figure 8) was risen significantly." This sentence is inconsistent with the chart.
  8. The overall structure of the paper is scattered, and the purpose of some results is not clear. It is suggested to strengthen the connectivity of the paper, grasp the key points for supplementary explanation, and omit unnecessary calculations and results.
  9. For Mongolian oak plants, this article studies the relationship between spring phenology and temperature changes, and concludes that the increase in temperature causes the green-up date to advance. How is this conclusion different from the conclusion of the previous study, in other words, can this conclusion be used as an innovative point of this research? The research conclusions need to be rearranged to find the innovation of the article.

Author Response

Reply to reviewer’s comments (922000, Round 1)

Reviewer 1

  • Why Mongolian oak was chosen as the research plant? What are the advantages of Mongolian oak compared with other plants? Have you compared the results of other vegetations? It is suggested to add a description of these contents, which can be put into the discussion section.

☞ We added the following contents in “2.1 ‘Study area’ subsection in “Materials and Methods” section. Lines 116 – 125.

The phenological signal of canopy reflectance from deciduous broadleaved forests can be clearly defined, ensuring accurate measurement of the change in growing season. Therefore, we selected Mongolian oak (Quercus mongolica Fisch.), which is not only a dominant species in the temperate deciduous broadleaved forest zone but also the representative species forming the late successional forest on the Korean Peninsula, as the target species of this study. Among the species belonging to the Quercus genus, Q. mongolica grows at the highest elevation and thus, would likely be a species sensitive to global warming. Mongolian oak begins leaf unfolding and senescence the earliest among deciduous oaks growing in Korea. Therefore, it is estimated that Mongolian oak could be the best species to derive phenological transition date from MODIS image with low spatial resolution.

  • Lines 110-111, why use "cherry blossom data" to verify "the change of green-up dates of Mongolian oak over year deducted from temperature variation"? It looks like two separate experiments. Why can we verify the correctness of the conclusion that Mongolian oak's green-up date is advanced by cherry blossom data? What is the relationship between the phenological phases of the two plants? What is the rationality of this verification method?

☞ We used the data to verify the change of green-up dates of Mongolian oak over years deducted from temperature variation because it is the data measured in field. Moreover, we have no the past data about Mongolian oak but we have the data for cherry blossom, which have recorded for about 100 years. In addition, the phenological data for cherry blossom, which have recorded for about 100 years, showed the close correlation with the temperature variation during the period. In this respect, this comparison is meaningful and significant.

  • The method part of the paper is too brief and the structure is chaotic. In section 2.2 Experimental design, the research data used was mentioned many times during the description of the experiment, but the description was not accurate enough to allow people to clearly know what data was used. And the calculation methods and processes of GDD, AGDD, AGDD threshold and green up dates should be introduced, even if the previous research methods are used, they should also be described in this paper. At the same time, it is suggested to add data processing diagram and experimental flow chart to make the research structure clearer.

☞ We revised our manuscript by accepting reviewer’s comments. Lines 173, 175 – 192. We also added Figure 2 by accepting reviewer’s comments.

  • Lines 139-141 refer to the 8-d interval air temperature map dataset. This data quotes the results of previous research. It is not described in detail here, including time resolution, spatial resolution, accuracy, etc. It is recommended to supplement the data description.

☞ We revised our manuscript by accepting reviewer’s comment. Lines 151 – 154.

  • Is there some repeats between the parts of section 2? It is suggested that when the experimental process and data are explained in detail, the repeated contents in each part should be removed, and the key points of each part should be highlighted. At the same time, the connection between the steps should be enhanced, which is helpful to strengthen the context connectivity.

☞ We revised “Materials and Methods” section of our manuscript by accepting reviewer’s comment.

  • English abbreviations appear in many places in the paper, such as "AGDD", "DoYAGDD", "DoY (159 AGDD)", "AGDD 159°C", etc. It is recommended to supplement the comparison table of proper noun abbreviations, and explain their representatives when they first appear The meaning and purpose of the research to enhance the reading experience of readers.

☞ We added abbreviation Box after Key words as the follows.

Abbreviations

GDD: Growing Degree Days set as 5°C

AGDD: Accumulated Growing Degree Days

EVI: Enhanced Vegetation Index (EVI)

MODIS LST: MODerate–Resolution Imaging Spectroradiometer Land Surface Temperature

DoYAGDD: Day of Year needed to reach the AGDD threshold (159 °C·d)

DoY (159 AGDD): Day of Year required to reach AGDD 159°C necessary for the leaf unfolding

AGDD 159°C: Accumulated Growing Degree Days 159 °C necessary for the leaf unfolding

  • There are some mistakes in the text, please check. Lines 255-256, "Flowering date of cherry (Figure 5) became earlier and air temperature (Figure 8) was risen significantly." This sentence is inconsistent with the chart.

☞ We revised the part.

  • The overall structure of the paper is scattered, and the purpose of some results is not clear. It is suggested to strengthen the connectivity of the paper, grasp the key points for supplementary explanation, and omit unnecessary calculations and results.

☞ We revised our manuscript by accepting reviewer’s comments.

  • For Mongolian oak plants, this article studies the relationship between spring phenology and temperature changes, and concludes that the increase in temperature causes the green-up date to advance. How is this conclusion different from the conclusion of the previous study, in other words, can this conclusion be used as an innovative point of this research? The research conclusions need to be rearranged to find the innovation of the article.

☞ We revised our manuscript by accepting reviewer’s comments.

Reviewer 2 Report

The authors have processed a large quantity of data and compressed it into several sets of figures and tables that almost clearly illustrate their points. 

It would be less deceptive and more meaningful to use the same scale for all sub-figures within a table. 

The authors are to be congratulated for preparing a well edited manuscript that is easy to read. Minor changes are suggested in the wording. This reviewer did not check the citations.

Author Response

Reply to reviewer’s comments (922000, Round 1)

Reviewer 2

The authors have processed a large quantity of data and compressed it into several sets of figures and tables that almost clearly illustrate their points. 

☞ Thank you for your kind encouragement.

It would be less deceptive and more meaningful to use the same scale for all sub-figures within a table. 

☞ We revised our manuscript by accepting reviewer’s comments.

The authors are to be congratulated for preparing a well edited manuscript that is easy to read. Minor changes are suggested in the wording. This reviewer did not check the citations.

☞ We revised our manuscript by accepting reviewer’s comments.

Reviewer 3 Report

The analyses carried out in the study are very simple and stronger conclusions could be achieved. Thus, the consideration of three different parts in the study is valuable but the integration of these three topics must be improved. Finally, the study of the effects of the lack of chilling accumulation in the phenology has been omitted in this study and could be relevant for some types of vegetation.

In addition, some specific concerns have been identified:

Title: The current title is too general. The evaluated systems must be included in the title.

Line 21. Explain DoYAGDD meaning.

Abstract: Although the three groups of results are described (Mongolian oak, cherry and cities vs. forest) but the abstract is confusing. The first and third one must be explained better including numerical results.

Introduction. For many crops, flowering requires fulfilment of two stages: endodormancy and ecodormancy. And then, in some cases, chill accumulation evaluation is required. Additional information about this term is required.

Mongolian oak has chill requirements? Describe. If these requirements are high, the flowering could be even delayed in the future. See studies as Wenden et al. 2019 in Global Change Biology.

Lines 89-90 Explain what is DoY (159 AGDD)

Lines 94-97. The correlation between Mongolian oak and cherry could be interesting. Have been reported chilling requirements for cherry?

Check the references number (for example Lim et al., [58])

Line 107. Which spatial resolution has the MODIS image?

Lines 124-135. Integrate these sentences in a single paragraph.

Lines 173-180. Explain better the meaning of normalized air temperature.

Lines 183-184. Additional explanations of deduction of green-up dates from EVI are required.

Lines 197-198. Could you provide additional information about the effect of urbanization on advancement of green-up?

Figure 2 (Left). The figure is difficult to see. Consider alternative map.

Line 208 and Fig. 3 “… MODIS-derived air temperature”: Which temperature? Mean air temperature in Spring? Explain better.

Fig. 3. Green-up start date observed or simulated? Please, explain.

Line 220. How was obtained EVI? Explain in the text.

Lines 220-222. What is the utility of these figures? Explain.

The comparison between AGDD DoY 2005 with WVI DoY 2005, etc. is required.

Line 225. How was measure this temperature?

Line 255. Please check the number of the figure.

Figs. 4, 5, 7 and 8. Range of temperatures (vertical axis) must be the same for each Figure in order to carry out comparisons.

Lines 257-258. The values for Mongolian oak were simulated while for cherry were observed. This must be highlighted.

Fig. 9. Were these values validated?

Lines 294-297. Which groups? Explain better, indicating at the beginning the analyzed groups.

Discussion section. The discussion section must integrate the different part of the study. Currently they are not integrated and then the conclusions are not clear.

The beginning of the discussion section must be a critical evaluation of the results obtained and following, they must be compared with previous one. The comparison of results must be funded in external studies (not previous studies with the same authors).

Check the numbers of the references

Lines 317-321. Were these digital cameras considered in this study? If not, please remove the information regarding them in Fig. 1. If they were considered, additional information is required in Materials and Method section.

Lines 322-336; 351-355. The discussion section is not a summary of the results; the reference to Figures and Tables must be exceptional.

Additional external studies must be included/evaluated/compared/discussed in the Discussion section, especially for the section 4.1.

Lines 356-364. This analysis is too long and it is based on empirical analyses of other authors and then, the results are too general. This paragraph is recommended to be removed.

Lines 391-393. Please remove this paragraph.

The conclusions repeat the results of the study. This section must be redone highlining the main conclusions obtained in the study.

What is the main utility of the results obtained in this study for an international reader? Could be extended to other crops or regions? Consider to provide some guidelines to carry out this extrapolation to other systems.

Author Response

Reply to reviewer’s comments (922000, Round 1)

Reviewer 3

The analyses carried out in the study are very simple and stronger conclusions could be achieved. Thus, the consideration of three different parts in the study is valuable but the integration of these three topics must be improved. Finally, the study of the effects of the lack of chilling accumulation in the phenology has been omitted in this study and could be relevant for some types of vegetation.

In addition, some specific concerns have been identified:

Title: The current title is too general. The evaluated systems must be included in the title.

Line 21. Explain DoYAGDD meaning.

☞ We explained the meaning in detain in ‘Materials and Methods’ as the follows.

Indices calculation

A series of indices that affect vegetation phenology were derived from the completely reconstructed 8-d air temperature maps. First, growing degree days (GDD; °C·d) were calculated using the equation from McMaster and Whilhelm (1997)[?]:

                                           (1)

where, Tmax·t, and Tmin·t are the maximum and minimum air temperatures at DoYt, respectively, and Tbase is the temperature below which plant growth is zero. In this study, we set the base temperature as 5°C (Gordon and Bootsma, 1993). In addition, the GDD based on meteorological station data were estimated to validate the GDD derived from MODIS.

We accumulated 8-d interval GDDs by simple summation when the GDD exceeded the base temperature (De Beurs and Henebry, 2004; Zhang et al., 2013):

GDDt > Tbase)                              (2)

where GDDt is the 8-d mean GDD at DoYt, and i is the time interval coefficient (GDD from MODIS : 8; GDD from field measured data: 1). AGDDt is the GDDs accumulated from the beginning of the time period until DoYt+7. Based on the average of AGDD threshold values determined from field measurements when the spring green-up started in Q. mongolica forests, a MODIS-derived AGDD threshold map was produced to assess the timing of green-up of Q. mongolica in all of South Korea.

Abstract: Although the three groups of results are described (Mongolian oak, cherry and cities vs. forest) but the abstract is confusing. The first and third one must be explained better including numerical results.

☞ Our study revealed that the phenology of the Mongolian oak is different by spatial temperature differences through satellite image interpretation, and based on the results, we tracked the change of the phenology of the Mongolian oak from the temporal temperature changes. An analysis of the temporal changes in the measured cherry blossom was carried out to compare these estimates. We contained these results in abstract.

Introduction. For many crops, flowering requires fulfilment of two stages: endodormancy and ecodormancy. And then, in some cases, chill accumulation evaluation is required. Additional information about this term is required.

Mongolian oak has chill requirements? Describe. If these requirements are high, the flowering could be even delayed in the future. See studies as Wenden et al. 2019 in Global Change Biology.

☞ As was referred in the paper, at warmer or oceanic climates, the beginning of the forcing period has been delayed, possibly due to insufficient chilling. In addition, the paper refers that in spite of a later beginning of the forcing period, spring phenology has continued to advance at these areas due to a faster satisfaction of heat requirements induced by climate warming. The purpose of the paper to reveal that ongoing climate warming will have different effects on the spring phenology of forest trees across latitudes due to the interactions between chilling, forcing and photoperiod.

But our study sites are located in the cool temperate zone of continental climate. Moreover, the Mongolian oak forest is usually established on the upper slope of the mountain. Therefore, there may be no effect of chilling.

Lines 89-90 Explain what is DoY (159 AGDD)

☞ We explained it in ‘Materials and Methods’

Lines 94-97. The correlation between Mongolian oak and cherry could be interesting. Have been reported chilling requirements for cherry?

☞ As we explained before, there may be no effect of chilling.

Check the references number (for example Lim et al., [58])

☞ We revised it.

Line 107. Which spatial resolution has the MODIS image?

☞ We added spatial resolution of the MODIS image in our manuscript. Line 117.

Lines 124-135. Integrate these sentences in a single paragraph.

☞ We integrated this part by accepting reviewer’s comment.

Lines 173-180. Explain better the meaning of normalized air temperature.

☞ We explained the meaning of normalized air temperature. Lines 215 – 217.

Lines 183-184. Additional explanations of deduction of green-up dates from EVI are required.

☞ We added explanations for a method to deduce green-up dates from EVI in our manuscript. Lines 220 - 227.

Lines 197-198. Could you provide additional information about the effect of urbanization on advancement of green-up?

☞ We explained the part additively using the map.

Figure 2 (Left). The figure is difficult to see. Consider alternative map.

☞ We revised the Figure (Figure 3, Figure number was changed from Figure 2 as Figure 2 was added newly).

Line 208 and Fig. 3 “… MODIS-derived air temperature”: Which temperature? Mean air temperature in Spring? Explain better.

☞ We explained the part in more detail. Figure 4 (Figure number was changed from Figure 3 as Figure 2 was added newly.

Fig. 3. Green-up start date observed or simulated? Please, explain.

☞ We obtained the dates from MODIS image.

Line 220. How was obtained EVI? Explain in the text.

☞ We explained the method in our manuscript. Lines 220 - 227.

Lines 220-222. What is the utility of these figures? Explain.

The comparison between AGDD DoY 2005 with WVI DoY 2005, etc. is required.

☞ Previously, other reviewers suggested that additional analysis is needed because the results of the year alone are less reliable. Therefore, we reinforced this by analyzing the correlation between the leaf unfolding date derived from AGDD in 2015 and the leaf unfolding dates derived from EVI in 2000, 2005, 2010 and 2015. In addition, we also analyzed the correlation between the leaf unfolding dates derived from EVI in 2000, 2005, 2010 and 2015. However, we are very sorry but time is some short.

Line 225. How was measure this temperature?

☞ We used data that the Korea Meteorological Administration provides.

Line 255. Please check the number of the figure.

☞ We revised it.

Figs. 4, 5, 7 and 8. Range of temperatures (vertical axis) must be the same for each Figure in order to carry out comparisons.

☞ We revised the figures to be the same size.

Lines 257-258. The values for Mongolian oak were simulated while for cherry were observed. This must be highlighted.

☞ We revised it.

Fig. 9. Were these values validated?

☞ Of course, we validated them.

Lines 294-297. Which groups? Explain better, indicating at the beginning the analyzed groups.

☞ We revised the part by referring groups composed of urban, rural, and forest areas.

Discussion section. The discussion section must integrate the different part of the study. Currently they are not integrated and then the conclusions are not clear.

The beginning of the discussion section must be a critical evaluation of the results obtained and following, they must be compared with previous one. The comparison of results must be funded in external studies (not previous studies with the same authors).

☞ We reinforced the following content in “Discussion” section.

The relationship between air temperature or degree days and phenophases, especially flowering and leaf unfolding is well known and has been widely reviewed (Cleland et al., 2007; 60). As temperature has been known as a major driver of phenology [25, 68], AGDD has a long history of use in predicting plant and insect phenology in agriculture [60, 69, 70].

Traditionally, the estimation of air temperature has depended on ground measurements at point levels such as meteorological stations and ground surveys. The ground-measured air temperatures were spatially interpolated using a GIS with conventional spatial statistics techniques such as kriging, inverse distance weighting (IDW), spline, and so on for expanding the estimation to the polygon level. Although the development of GIS and spatial statistics have led to a drastic refinement in the interpolated result, there is a severe weakness due to the limited number of the points [60,71].

But remote sensing is an alternative data source since remotely sensed imagery is intrinsically spatialized [59]. In particular, MODIS provides an abundant series of LST (land surface temperature) products with different spatial and temporal resolutions from both Terra and Aqua platforms [57]. Previous studies have shown that the LST data measured by MODIS can be successfully used for linear regression estimates of air temperatures at a regional scale [60,72,73,74].

Check the numbers of the references

☞ We checked thoroughly and revised the reference numbers.

Lines 317-321. Were these digital cameras considered in this study? If not, please remove the information regarding them in Fig. 1. If they were considered, additional information is required in Materials and Method section.

☞ We didn’t include data obtained from digital camera. So, we revised our manuscript by accepting reviewer’s comment. Figure 1.

Lines 322-336; 351-355. The discussion section is not a summary of the results; the reference to Figures and Tables must be exceptional.

☞ We reinforced “Discussion” section by accepting reviewer’s comments.

Additional external studies must be included/evaluated/compared/discussed in the Discussion section, especially for the section 4.1.

☞ We reinforced “Discussion” section by accepting reviewer’s comments.

.

Lines 356-364. This analysis is too long and it is based on empirical analyses of other authors and then, the results are too general. This paragraph is recommended to be removed.

☞ We removed the part.

Lines 391-393. Please remove this paragraph.

☞ We removed the paragraph.

The conclusions repeat the results of the study. This section must be redone highlining the main conclusions obtained in the study.

What is the main utility of the results obtained in this study for an international reader? Could be extended to other crops or regions? Consider to provide some guidelines to carry out this extrapolation to other systems.

☞ We revised “Conclusion” section drastically by accepting reviewer’s comments. Lines 494 – 522.

This manuscript is a resubmission of an earlier submission. The following is a list of the peer review reports and author responses from that submission.

Round 1

Reviewer 1 Report

Hi,

I have annotated my comments in the attached document. However, one of the thing that keeps coming up is whether the authors made the study on Mangolian Oak or Cherries or both? In one paragraph, they give the impression that Mangolian Oak is their object of study. Immediately, in the next they bring Cherries into the picture. This could be made clearer.

Without equations 1& 2, suddenly equations 3&4 appear, with no explanation of what they are going be used for? Anyone would guess it for the section 3.3. However, no mention of this equation was made about these equations were made in the section. 

No clear description on the methodology followed for the study.

No plots or figures, justifying the methodology/equations.

Hence, i believe this paper has to be rewritten mostly with clear objective, clear motivation, and methodology along with justifications.

Thanks 

Author Response

Reviewer 1

I have annotated my comments in the attached document. However, one of the thing that keeps coming up is whether the authors made the study on Mongolian Oak or Cherries or both? In one paragraph, they give the impression that Mongolian Oak is their object of study. Immediately, in the next they bring Cherries into the picture. This could be made clearer.

☞ We revised our manuscript by accepting reviewer’s comment. Our study was focused on Mongolian oak. In addition, we also analyzed a phenology event of cherry based on the data obtained from the meteorological stations to reinforce the results that we collected through the investigation for Mongolian forest.

Without equations 1& 2, suddenly equations 3&4 appear, with no explanation of what they are going be used for? Anyone would guess it for the section 3.3. However, no mention of this equation was made about these equations were made in the section. 

No clear description on the methodology followed for the study.

No plots or figures, justifying the methodology/equations.

Hence, I believe this paper has to be rewritten mostly with clear objective, clear motivation, and methodology along with justifications.

☞ We revised our manuscript by accepting reviewer’s comment.

Thanks 

Reviewer 2 Report

Manuscript ID remotesensing -669445#  Monitoring for changes in spring phenology at both temporal and spatial scales based on MODIS LST data in South Korea #

Here are my comments

The MS does not have the recent citation from the related investigations. For example, authors cited ONLY one investigation for 2019, while they did NOT cite any investigation published in 2018 or 2017. Thus, authors must read the recent investigations and cite them where it is possible within the MS. There is no clear statistical analysis methods used in the data of MS. Why? Authors must write what statistical analysis methods were used for their data. Abstract is good, but it will be better if the authors added some values/data within the abstract. Introduction is ok, and the aims are clear. However, it will be better to include the hypothesis of this study. Methods should be written in details. Statistical analysis part in M&M should be written in a separated paragraph. Results is well written. Discussion must be improved and further citations are needed. English language and style are fine/minor spell check required. Conclusion is well understandable, but it has to be shorter than the current version.

Author Response

Reviewer 2

Here are my comments

The MS does not have the recent citation from the related investigations. For example, authors cited ONLY one investigation for 2019, while they did NOT cite any investigation published in 2018 or 2017. Thus, authors must read the recent investigations and cite them where it is possible within the MS.

☞ We revised our manuscript by citing recently published papers.

There is no clear statistical analysis methods used in the data of MS. Why? Authors must write what statistical analysis methods were used for their data.

☞ We revised our manuscript by adding “Statistical analysis” section.

Abstract is good, but it will be better if the authors added some values/data within the abstract.

☞ We revised our manuscript by accepting reviewer’s comment.

Introduction is ok, and the aims are clear. However, it will be better to include the hypothesis of this study.

☞ We revised our manuscript by accepting reviewer’s comment.

Methods should be written in details. Statistical analysis part in M&M should be written in a separated paragraph.

☞ We revised our manuscript by accepting reviewer’s comment.

Results is well written.

Discussion must be improved and further citations are needed. English language and style are fine/minor spell check required.

☞ We revised our manuscript by accepting reviewer’s comment.

Conclusion is well understandable, but it has to be shorter than the current version.

☞ We revised our manuscript by accepting reviewer’s comment.

Reviewer 3 Report

The authors demonstrate the correlation between increasing air temperature and onset of the spring green up of Mongolian oak in South Korea.

The paper is potentially of interest in demonstrating this correlation. However, the role of remote sensing is rather marginal, as the correlation is fully explained from in situ measurements at weather stations and observations on spring phenology. MODIS land surface temperature (LST) is used to spatialize air temperature, but it is not evident why this needs to be done, given the dense network of weather stations and phenology observations that are relatively close to these stations. An effort to determine spatial patterns in start-of-season with, for instance, MODIS EVI, and for some of the extremes in the period 1990-2019, would have been a more suitable remote sensing use case.

Overall, the quality of the manuscript is high. Ordering of the material can be improved. Language-wise, there are only few issues.

Details:

Abstract

L 25: "responded negatively" in unclear. If the decreasing trend in DoY of flowering is meant, this is already clear from the first part of the sentence.

L 27: "Applied the result..." should be "Extrapolating the result.

L 28: "with the rise of air temperature due to global warming". The latter is not proven in this paper, as results only relate to South Korea.

L 31-33: the authors only expand spatiotemporally the pattern in air temperature, not that of spring vegetation phenology. No satellite observations on spring vegetation phenology are made, only point observatiions. This is a major weakness in the paper.

Introduction

L 43. "the lifecycle..." should be "the seasonality".

L. 44. ... which is related to recent climate change" is not relevant here, because already stated earlier. "spring phenology" is a monitoring tool per se, with or without climate change.

L . 90 "usually" should be "mostly" (lowlands do not change location)

L 107: unclear sentence

L 112: should be "where local microclimatic variation can be high". "very severe" is too strong, and not evident in Fig 7, apart from the UHI effects.

L 115: best to use a single, consistent format for dates (the one on line 109)

L 122: provide reference for FTP server

L 124: more likely WGS84

Section 2.3: it is unclear how MODIS Land Surface Temperature is translated into air temperature. It is unclear what is measured and normalized data. GDD normally has a baseline temperature (eg. 3 C), under which they do not contribute to AGDD. This is not given. Equations (3) and (4) are not well explained.

Results

Weather data is only available from 1 Jan - 2 Jul 2015 (DoY 1-184). How are "winter", "spring" and "summer" periods defined?

Fig 2: this figure can also be derived from measured air temperature at the nearest weather station, avoiding the need for MODIS LST. This should be relavant to add.

Fig 3: unclear what green up is plotted here? Apparently not Mongolian Oak?

Fig 5: caption should be "Simulated green-up dates of... requiring 159 AGDD and using the measured air temperature trend etc". [some readers may only scan figures, and it should be clear that these are not real observations).

L 211. ...experiences thermodynamic due... is unclear. "Temperature gradients"?

Fig 7. Do "temperature anomalies" relate to Eq (3) and (4)?

L 251: rose between 0.9 to 2.5

L 259-262: can these increases be expressed relative to total annual GPP/NEP as well?

L 278 proportionated -> attributed

L 291-297. This is incorrect. Satellite imagery has only played a role to establish the temperature pattern (which was not really required). The trends in vegetation phenology has not been cross-analysed with satellite data. This should be rephrased.

L 311 "The result comes from that" unclear.

L 310-315. This paragraph needs rephrasing. Something along the line "the air temperature increases over the last century are especially significant in UHI. This has led to higher AGDD and, therefore, higher risks of phenological mismatch.". The problem is that the authors have only proven that UHI temperatures are higher than in surrounding rural and forest areas. It is still unknown whether the increase over the last century in the non-urban classes has been different (is this because only the urban weather stations have a 100 year registration history??).

A major revisions is required to address the point above.

Author Response

Thanks. 

Round 2

Reviewer 1 Report

Dear Authors, 

Please accept my sincere thanks for quick turn around. 

Please find attached my comments in the attached annotated manuscript. 

My major concern is that as per your manuscript, there are different land cover classes and land use cases. Similarly, you are considering two different species of plants. In this case, you have to adopt a good sampling strategy that is statistically and logically justified. This is lacking in this manuscript. Nor have you attempted to provide any statistics on them or explain them with plots and graphs. In this case, the conclusions and results become skewed and becomes really difficult to understand the figures.

Without a such a systematic approach, I believe that manuscript is not ready to be published. Hence, I would like to urge the authors to adopt a systematic scientific approach with scientific justification of the steps in the methodology and resubmit the manuscript.

Thanks and Regards. 

Reviewer 2 Report

MS looks better than before.